# Application of Curcumin Emulsion Carrier from Ultrasonic-Assisted Prepared Octenyl Succinic Anhydride Rice Starch

**DOI:** 10.3390/molecules27206955

**Published:** 2022-10-17

**Authors:** Yuxue Zheng, Huiling Zhang, Xiaobo Wei, Haitian Fang, Jinhu Tian

**Affiliations:** 1Ningxia Key Laboratory for Food Microbial Applications Technology and Safety Control, Ningxia University, Yinchuan 750021, China; 2College of Biosystems Engineering and Food Science, National-Local Joint Engineering Laboratory of Intelligent Food Technology and Equipment, Zhejiang Key Laboratory for Agro-Food Processing, Integrated Research Base of Southern Fruit and Vegetable Preservation Technology, Zhejiang International Scientific and Technological Cooperation Base of Health Food Manufacturing and Quality Control, Zhejiang University, Hangzhou 310058, China; 3Fuli Institute of Food Science, Zhejiang University, Hangzhou 310058, China; 4Ningbo Research Institute, Zhejiang University, Ningbo 315100, China; 5Food and Healthy Researcher Center, Zhejiang University Zhongyuan Institute, Zhengzhou 450000, China

**Keywords:** rice starch, octenyl succinic anhydride, curcumin, emulsion

## Abstract

The emulsification of ultrasonic-assisted prepared octenyl succinic anhydride (OSA) rice starch on curcumin was investigated in the present study. The results indicated that the encapsulation efficiency of curcumin in emulsions stabilized by OSA-ultrasonic treatment rice starch was improved, from 81.65 ± 0.14% to 89.03 ± 0.09%. During the in vitro oral digestion, the particle size and Zeta potential of the curcumin emulsion did not change significantly (*p* > 0.05). During the in vitro digestive stage of the stomach and small intestine, the particle size of the curcumin emulsion continued to increase, and the absolute potential continued to decrease. Our work showed that OSA-pre-treatment ultrasonic rice starch could improve curcumin bioavailability by increasing the encapsulation efficiency with stronger stability to avoid the attack of enzymes and high intensity ion, providing a way to develop new emulsion-based delivery systems for bioactive lipophilic compounds using OSA starch.

## 1. Introduction

Curcumin is a natural small molecule polyphenol compound with anti-inflammatory, anticancer, and antioxidant functions [1,2]. It also has strong anticorrosion ability and does not produce toxic side effects. It is often used as a natural food preservative, colorant, and antioxidant in beverage and food [3]. However, its high lipophilicity limits its absorption in the human body, and its bioavailability is reduced due to its poor water solubility, high metabolic degradation rate, and easy removal from the body during biotransformation [4]. To solve the above problems, a large number of studies have used different delivery systems to encapsulate curcumin, such as hydrogel, nanoparticles, liposomes, etc. [5]. Among them, a curcumin delivery system based on emulsion is the most important research direction, as it can improve the bioavailability and permeability of curcumin and reduce the metabolic degradation in the process of biotransformation [6].

Natural starch has poor solubility in cold water, thermal instability, easy coagulation, and poor film forming and emulsification. In order to better meet the requirements of industrial production, starch is usually modified by physical, chemical, or biological methods. Octenyl succinate starch ester (OSA starch) is a widely used chemically modified starch. It is prepared by esterification of octenyl succinic anhydride (OSA) and starch under alkaline conditions. The introduction of hydrophobic groups (olefin long chain) and hydrophilic groups (carboxyl group) makes OSA starch amphiphilic [7]. Attributing to its good solubility in cold water, high biosafety, strong emulsification and embedding ability, and stable resistance to pH and other environmental factors, OAS starch has attracted extensive attention amongst researchers [8].

Ultrasonic treatment influences the starch particle morphology and molecular structure through microjet, mechanical shear, and local heating generated by the cavitation mechanism [9]. At the same time, free radicals generated at the moment of cavitation bubble collapse will also attack the molecular structure of starch, making starch more prone to chemical modification [10]. Ultrasonic chestnut starch can also enhance the encapsulation efficiency and bioactivity of resveratrol [11]. At present, the research on OSA starch assisted by ultrasonic technology mainly focuses on corn starch and cassava starch. Zhang et al. [12] first applied 500 W ultrasound to treat corn starch, and then reacted the ultrasonic pre-treated starch with OSA and found that the mechanochemical action of ultrasonic treatment significantly changed the apparent morphology and crystal structure of corn starch. However, whether ultrasonic pre-treated rice starch with OSA has better emulsifying ability and how efficient it is at delivering curcumin is still unknown [13].

In order to explore the effects of different properties of OSA rice starch assisted by ultrasound on the efficiency of curcumin delivery, we used ultrasonic-assisted OSA rice starch as the emulsifier to construct the curcumin emulsion carrier. The loading rate and bioavailability of different emulsions were measured, and the particle size, Zeta potential, apparent morphology, and free fatty acid release of the emulsion at different in vitro digestive stages were observed and further established the way and mechanism of how ultrasonic-assisted OSA rice starch affects the release of curcumin during in vitro digestion Therefore, the present study might provide as a reference for the development of the potential application of ultrasonic-assisted OSA rice starch on the encapsulation of bio actives.

## 2. Results and Discussion

### 2.1. Loading Efficiency/Capacity of Curcumin in OSA Rice Starch

As shown in Table 1, compared with the native rice starch emulsion, ultrasonic rice starch, especially UORS-600 W, had the highest degree of substitution (DS) and replacement of efficiency (RE) (*p* < 0.05), indicating that higher intensity ultrasonic treatment had more serious damage to the starch structure, and the amorphous area in the starch and even part of the crystallization area was also affected. Therefore, OSA substitution not only occurred in the amorphous area on the surface of the starch particles, but also reacted with the active site in the crystallization area, resulting in an increase of DS and RE [14].

The encapsulation efficiency of curcumin in the OSA rice starch emulsion system increased by 0.97–1.15 times, while the loading capacity increased from 1.36 µg/mg to 1.81 µg/mg (Table 1), indicating that OSA rice starch was more efficient at loading curcumin, which was consistent with the study by Pan et al., who found that OSA-modified dextrin improved the encapsulation efficiency of curcumin with the increasing ultrasonic power [15]. According to a previous study, the encapsulation efficiency increased [15]. This phenomenon may be attributed to the larger DS and more hydrophobic groups on the surface of OSA rice starch prepared by ultrasonic-assisted 600 W. In a study of OSA dextrin-curcumin emulsion, it was also found that with the increase of DS, the hydrophobic interaction between the molecules was enhanced, and the loading efficiency increased. However, when the OSA content exceeded 10%, the loading rate no longer increased, probably because when the OSA concentration exceeded a certain level, an excessive alkenyl chain would wrap the dextrin in the center, impeding the interaction between the oil phase and the hydrophobic molecular chain [16,17].

### 2.2. Particle Size during the In Vitro Digestive Process

According to Figure 1a, before simulated digestion, the particle size of the OSA rice starch emulsion that was assisted by ultrasound was smaller than that of the OSA starch group and native starch group when using the traditional water phase method (*p* < 0.05). After simulated oral digestion, the particle sizes of all emulsions increased slightly. Compared with the initial emulsion, the particle sizes of NRS, ORS, UORS-150W, and UORS-600W increased by 33%, 25%, 8.1%, and 7.7%, respectively. However, compared with other digestive stages, the change of particle size in oral digestion was not obvious (Figure 1a). α-amylase in oral fluid slightly hydrolyzed OSA starch on the surface of the emulsion drops, leading to droplet coalescence and flocculation. However, due to the oral digestion time of only 3 min, the effect on the droplet size was not obvious. This indicated that the OSA starch stabilized emulsion could effectively resist the attack of high ionic strength and α-amylase in the saliva, and maintained a stable particle size and particle size distribution.

After digestion in the stomach, the particle size of all the emulsions increased significantly. Since there was no corresponding enzyme in the gastric digestive fluid to hydrolyze starch, it was speculated that the change of particle size was mainly due to the higher ionic strength and lower pH that destroyed the electrostatic balance of the emulsion system [18]. It was found that the NRS particle size changed the most, from 1159.50 ± 242.54 nm to 15,045.50 ± 1040.15 nm, while the ORS particle size increased from 1015.83 ± 98.94 nm to 3344.67 ± 313.7 nm. This result indicated that the emulsion formed by OSA rice starch was more stable in the gastric digestion stage [19]. However, ultrasonic assistance could further alter the emulsion particle size (the particle size of UORS-150W and UORS-600W increased by 1.85 and 1.10 times, respectively), which might be attributed to the larger DS of OSA starch prepared by ultrasonic assistance. Therefore, the ultrasound-assisted OSA starch emulsion showed higher stability, mainly because of the larger DS and the closer emulsified layer structure that was formed by the starch molecular chain on the surface of the droplet, which improved the stability of the emulsion in the digestive process of gastric fluid [20,21].

After 5 min of simulated intestinal digestion, the particle size of latex continued to increase, in which the particle size of the NRS, ORS, UORS-150W, and UORS-600W emulsion increased by 34.43%, 29.46%, 9.88% and 16.67%, respectively. A similar phenomenon was also reported in a study of an OSA nanoemulsion loaded with lycopene [16]. It was inferred that the rapid increase of the emulsion particle size was due to the hydrolysis of a large amount of OSA starch by pancreatic amylase and glycoside enzymes in the intestinal fluid. The emulsion structure was seriously damaged. At the same time, bile salts abundant in small intestinal fluid would also adsorb on the surface of the emulsion droplets, replacing the original emulsifier and leading to droplet coalescence [22]. By comparing the degree of particle size increase of the emulsion of different components, the ultrasonic assisted OSA starch stabilized emulsion had a smaller particle size change and more stable properties after intestinal digestion for 5 min, which might be related to the strong anti-digestibility of OSA starch. It was speculated that the higher stability of the OSA starch emulsion in the small intestine might be attributed to the higher DS, the strong anti-digestion ability, and the more effective resistance to amylase hydrolysis. When intestinal digestion reached 120 min, the particle size of all samples decreased sharply to lower than that of 100 nm, and the oil in the emulsion system was almost completely digested [23].

### 2.3. Zeta Potential Variation during the In Vitro Digestive Process

As shown in Figure 1b, the initial potential range of all starch emulsions before digestion ranged from −24.43 to −16.50 mV. During the simulated digestion process, the absolute value of emulsion Zeta potential showed a peak trend, which was similar to the results of previous studies [24]. After oral digestion, the Zeta potential was negative, and the negative charge in the emulsion system mainly came from the carboxyl group (-COO-) of the OSA starch. The absolute value of the Zeta potential decreased slightly, which might be due to α-amylase in oral fluid hydrolyzing part of the starch and disturbing the balance of the emulsion structure. At the same time, a large number of mineral ions contained in the emulsion would also adsorb on the surface of the droplet and produce an electrostatic shielding effect [25]. However, the time of digestion in the oral part of the emulsion was short, so the change of the Zeta potential was not significant.

In the gastric digestion stage, the absolute value of emulsion Zeta potential decreased significantly, indicating that the charge density on the droplet surface decreased. It was noteworthy that there was no significant difference in the Zeta potential between NRS and ORS in the gastric stage (*p* > 0.05), but a significant difference in the particle size between the two components. The results showed that although the increase of the surface charge of the emulsion droplets could enhance the electrostatic repulsion between the droplets, the absolute value of the Zeta potential of the emulsion samples at the stomach digestion stage was at a low level, which was not enough to maintain the stability of the emulsion, and the stability of the emulsion depended on the steric hindrance. The results could explain why the ultrasound-assisted OSA starch emulsion maintained higher stability at the stomach stage: on the one hand, the ultrasound-assisted OSA starch had larger DS, which could increase the surface charge density of the emulsion drops and enhance the electrostatic repulsion between the emulsion drops. On the other hand, the starch molecules are connected with more OSA groups, and the OSA groups are more evenly distributed in starch, which is conducive to the formation of the emulsified layer, with strong rigidity and a tight structure, and can better protect oil droplets.

After 5 min of intestinal digestion, the absolute value of the Zeta potential of all emulsions reached the lowest point, indicating that the emulsion structure was further destroyed. The Zeta potential of the latex prepared by original rice starch was −3.00 ± 0.14 mV, and that of the latex formed by OSA starch prepared by the traditional aqueous method was −6.87 ± 0.49 mV. However, the Zeta potential of the OSA starch latex prepared by ultrasonic assisted preparation was greater than −12 mV. This indicated that the ultrasonic-assisted OSA starch group could better maintain the structure of emulsion.

After 120 min of intestinal digestion, the absolute potential value of all OSA starch emulsions increased and even exceeded the level before simulated digestion, and the absolute potential value of OSA starch emulsions assisted by ultrasound was significantly higher than that of other components (*p* < 0.05). It was speculated that the absolute potential of the OSA starch group with ultrasonic assistance was higher, which might be because more free fatty acids were generated by hydrolysis during intestinal digestion [26].

### 2.4. The Change of the Emulsion Microstructure during the Digestive Process

In order to investigate the dynamic changes of the micromorphology of emulsions formed by different starches during the simulated digestion, starch and oil in emulsions were stained with Nile blue and Nile red, respectively, and the morphological changes of emulsions during the different simulated digestion stages were observed by a laser confocal microscope. As shown in Figure 2 (full resolution images of CLSM can be obtained from the Appendix A), the oil droplets showed red fluorescence, and the starch showed green fluorescence. The part showing yellow fluorescence was formed by the superposition of oil droplets and starch.

As could be seen from Figure 2A1, before simulated digestion, all emulsion samples showed small droplets with a small particle size. After oral digestion, a small number of aggregated large oil droplets appeared in the original rice starch group (Figure 2A2), while no significant morphological change in the OSA starch group was observed.

In the stomach stage, the aggregation of emulsion droplets was intensified in all samples, among which, the aggregation degree of droplets in the original rice starch group was larger, and larger flock-like or even block-like aggregates appeared. However, the ultrasound-assisted OSA starch emulsion could still maintain a small and evenly distributed microstructure. It was also found that the original rice starch group showed more obvious red/orange fluorescence, while the ultrasonic-assisted OSA starch group showed more obvious light orange/yellow fluorescence, indicating that the surface of oil droplets in ultrasonic-assisted OSA starch emulsion was covered with more starch.

After 5 min of intestinal digestion, large oil droplets formed by droplet coalescence appeared in the original rice starch group, and the droplet color was dark red, indicating that the starch-based emulsifier on the surface of the oil droplets was almost completely hydrolyzed, and the decomposition of the surface emulsifier would lead to serious aggregation of the emulsion. The droplet sizes of the OSA starch group also increased to varying degrees. The order of droplet sizes was ORS > UORS-150W > UORS-600W, which was consistent with the particle size results determined in Section 2.3. By comparing the color of the CLSM images, it was found that the droplets in the ORS group were orange-red, and there was a light green interface film on the edge of the oil droplets, which represented OSA starch covering the oil droplets. With the increase of ultrasonic assisted intensity, the red oil droplets in CLSM images gradually decreased, indicating that the degree of oil hydrolysis increased, which might attribute to the smaller emulsion particle size [27]. However, after 120 min of small intestine digestion, there was almost no red color in the CLSM images, indicating that the oil had been completely digested.

Previous studies also found that after OSA starch emulsion was digested in the small intestine, the number of oil droplets decreased, which might relate to the following factors: (1) the small intestine digestive liquid added would dilute the emulsion system; (2) Ca^2+^ forms a complex with cholate; (3) pancreatic lipase hydrolyzed the oil; (4) the hydrolysates of oil were compounded with particles, vesicles, and micelles in the emulsion system [28]. At the same time, with the increasing ultrasonic assisted power, the yellow color of the CLSM image became more obvious, indicating that more undigested OSA starch was left in the emulsion. The presence of starch-based emulsifiers could provide a protective barrier for the oil droplets, thus improving the stability of the emulsion structure [29].

### 2.5. Free Fatty Acid Release Rate

As shown in Figure 3, free fatty acid (FFA) release curves of all samples rose rapidly at first, then slowed down, and finally approached a constant value, which was consistent with previous research results [30]. The release rate of FFA was determined by the slope of the curve during the rapid digestion phase (0–20 min), while the final release of FFA was determined by the constant value finally achieved. The order of the FFA release rate of the emulsion samples is UORS-600W > UORS-150W > ORS > NRS, which corresponded to the phenomenon that the number of oil droplets in OSA starch emulsion assisted by ultrasound decreased the fastest. The order of FFA final release was slightly different, as UORS-150W (67.99%) and UORS-600W (67.34%) were greater than NRS (42.73%) and ORS (60.22%), but no significant difference between the two ultrasonic-assisted components were observed (*p* > 0.05). The above results indicated that the digestion rate and degree of oil in OSA starch emulsion prepared by ultrasound were improved, which might have attributed to the smaller particle size of OSA starch emulsion prepared by ultrasound. A large number of studies have confirmed the relationship between the oil digestion rate and oil droplet interface area. The smaller the droplet diameter was, the larger the effective interface area of oil and enzyme contact would be, thus improving the oil digestion rate [31,32,33]. Based on the above discussion about emulsion particle size and microstructure changes, it was speculated that OSA starch assisted by ultrasound had higher DS and good emulsification performance, which was beneficial to form a starch-based emulsification layer with strong rigidity and high compactness, and inhibited the flocculation and aggregation of oil droplets in the gastrointestinal tract. At the same time, the higher DS reduced the degree of starch digestion, maintained the emulsifier on the surface of oil droplets, improved the structure stability of emulsion during the simulated digestion process, and finally led to a decrease of the emulsion particle size and an increase of the oil digestion rate.

### 2.6. Bioavailability of Curcumin in the Emulsion

Only when the lipid soluble functional factors in the emulsion system were released from the carrier matrix into mixed micelles or vesicles could they be absorbed and transported by the small intestinal epithelial cells. The proportion of lipid soluble functional factors transferred to micelles, namely bioavailability, is an important indicator that is widely used in in vitro digestion experiments.

As can be seen from Table 2, the bioavailability of curcumin in the emulsion carrier system constructed by original rice starch is low, only 10.44%, while the bioavailability of curcumin in the carrier system stabilized by OSA rice starch is significantly increased. UORS-150W (23.97%) and UORS-600W (23.48%) were higher than ORS (20.47%), but there was no significant difference in the bioavailability between the two ultrasonic-assisted OSA starch groups (*p* > 0.05).

A positive correlation was found between the bioavailability of the curcumin emulsion and the final release of free fatty acids (FFA) [27]. Studies have demonstrated the release and absorption of lipid-soluble bioactive substances in vitro to simulate intestinal digestion: first, oil droplets are hydrolyzed under the action of enzymes to release the curcumin dissolved in them; curcumin is then self-assembled into mixed micelles or vesicles consisting of lipid digestion products (such as free fatty acids and glycerol) and cholates, for subsequent absorption by simple diffusion by the small intestinal epithelial cells [34]. Therefore, it is speculated that the digestion rate of oil affects not only the release rate of curcumin from oil droplets, but also the formation of oil hydrolysates and the content of curcumin that can be absorbed and utilized in mixed micelles.

## 3. Materials and Methods

### 3.1. Samples and Chemicals

Rice starch (CAS: S7260), curcumin (CAS:458-37-7), pepsin (CAS: 9001-75-6), porcine pancreatic α-amylase (CAS: 9000-90-2), α-amyloglucosidase (106 U/mL, CAS:9032-08-0), invertase (300 U/mg, CAS: 9001-57-4), and bile (CAS: 8008-63-7) were purchased from Sigma-Aldrich. Soy oil was bought from Yihaikerry Co., Ltd. (Shanghai, China). Octenyl succinic anhydride and other chemicals of analytic grade were purchased from Aladdin Co., Ltd. (Shanghai, China).

### 3.2. Preparation of OSA-Rice Starch

The ultrasonic rice starch was prepared according to Yang et al. [10]. Briefly, 35% (*w*/*w*) starch slurry was treated with ultrasonic powers 150 W and 600 W with an ultrasonic device (JY92-IIDN, Xinzhi, China,) under 25 °C for 20 min and then dried at 40 for 24 h. Native rice starch, rice starch with ultrasonic powers 150 W and 600 W in 25 °C were labeled as RS, URS-150W, and URS-600W, respectively. Ultrasonic pre-treated rice starch was esterified with OSA using the method of Zhang et al. [14]. The starch suspension was briefly dispersed with distilled water (35%, *w*/*w*) and the pH was adjusted to 8.5. Then, OSA (3%, on the basis of dry starch) was added slowly within 1 h, with continuous stirring at 35 °C, while maintaining a pH at 8.5 and reacting for 2 h. The reaction was terminated by adjusting the pH to 6.5 and the mixture was centrifuged at 8000 rpm for 10 min. The sediment was washed twice with distilled water and then twice with ethanol to remove any free OSA. The OSA-starch samples were oven-dried at 40 °C for 24 h and passed through a 100-mesh standard sieve. For reference, native rice starch without ultrasonic treatment was also modified with OSA, as described above. OSA-starch samples prepared by the conventional aqueous method and by ultrasonic pre-treatments under different powers were named as ORS, UORS-150W and UORS-600W, respectively. The native rice starch was also labelled as NRS.

### 3.3. Emulsion Preparation

A total of 5.0% (*w*/*w*) of NRS, ORS, UORS-150W, and UORS-600W was gelatinized for 20 min and then cooled to room temperature for the water phase. A total of 0.2% (*w*/*w*) curcumin was dissolved in soy oil and stirred in room temperature for 1 h for the oil phase. Subsequently, the mix ratio of the water phase and the oil phase was 4:1 (*w*/*w*), and then the crude emulsion was prepared by high-speed shearing for 3 min with a high-speed disperser (FSH, Jintan, China) at 15,000 rpm/min. The final curcumin carrier emulsion system was obtained by using a high-pressure homogenizer (Nanogenizer-30K, Genizer, Irvine, CA, USA) with a pressure of 150 MPa. The degree of substitution (DS) and reaction efficiency (RE) of OSA-starch were measured using the titration method [16].

### 3.4. Curcumin Loading Rate

A series of different concentrations of curcumin standard solutions (0, 1, 2, 3, 4, 5, 6, and 7 μg/mL) were prepared with ethanol. The absorbance of all curcumin solutions was read with 425 nm with a spectrophotometer (UV-2600, Shimadzu, Japan) and the standard curve was made. One mL emulsion with 5 mL dichloromethane was stirred thoroughly for 2 min, and the downward layer was collected, and the absorbance was read at λ = 425 nm. The free curcumin in the emulsion was calculated using Equation (1).
Encapsulation efficiency (%) = (1 − free amount of curcumin/total amount of curcumin) × 100%(1)

### 3.5. Simulated Digestion

The simulated digestion process containing the oral, gastric, and intestinal phases was used to monitor the potential digestion of all samples and the corresponding fluid was prepared as described previously—with minor modifications [17]. The droplet size, zeta potential, microstructure of the emulsion, and the released rate of free fatty acid were investigated.

Stimulated saliva fluid: NaCl (0.298 g), KCl (0.896 g), NaH_2_PO_4_ (0.888 g), NaHCO_3_ (1.694 g), Na_2_SO_4_ (0.57 g), KSCN (0.2 g), and (NH_4_)_2_CO_3_ (0.2 g) were dissolved in 1000 mL deionized water and the pH was adjusted to 6.8. Emulsions were mixed with SSF at a volume ratio of 1:1, and the pH was adjusted to 6.8. The mixtures were continuously stirred at 37 °C for 3 min at 100 rpm.

Stimulated gastric fluid: NaHCO_3_ (2.1 g), NaCl (2.758 g), MgCl_2_ (0.1 mM), KCl (0.515 g), KH_2_PO_4_ (0.122 g), and (NH_4_)_2_CO_3_ (0.048 g) were dissolved in 1000 mL deionized water, and the pH was adjusted to 2.5 with HCl. Before use, pepsin (final 2000 U/mL) was added to the solution. The sample from the oral phase was mixed with the SGF at a volume ratio of 1:1, and the pH was adjusted to 2.5. The mixtures were continuously stirred at 37 °C for 2 h at 100 rpm.

Stimulated intestinal fluid: CaCl_2_ (0.244 g), bile salt (44 g), and pancreatin (2.2 g) were dissolved in 1000 mL deionized water and the pH was adjusted to 7.0. Before use, bile salts (10 mM) and pancreatin (1:3000) were added to the solution. The sample from the gastric phase was mixed with the SIF at a volume ratio of 1:1, and the pH was adjusted to 7.0. The mixtures were continuously stirred at 37 °C for 2 h at 100 rpm.

### 3.6. In Vitro Bioavailability of Curcumin

After simulated in vitro digestion, the digestive solution was centrifuged at 10,000 *g*, 4 °C, for 40 min. One mL raw digestive solution from the intestinal phase was mixed well with 5 mL chloroform/ethanol (3:2, *v*/*v*). the organic layer containing curcumin was collected, and the mixture was extracted twice. The extraction solution was combined and diluted with ethanol. The UV absorbance value was measured at 425 nm and the concentration of curcumin was calculated according to the standard curve. The formula for calculating the bioavailability of the emulsion system is as follows:Bioavailability (%) = Concentration of curcumin in organic layer/concentration of curcumin in emulsion × 100%.(2)

### 3.7. Particle Size and Zeta Potential Measurement

The particle size and Zeta potentials of the starch sample were characterized using the static light scattering instrument (Beckman Coulter LS230, Coulter Co., Pasadena, FL, USA). Before analysis, the initial, oral, stomach, and small intestinal (5 min and 120 min) diluted with deionized water was used as dispersant, and the suspension was sonicated for 5 min to disperse the starch granules before testing. The refractive index of water and starch was set as 1.33 and 1.50, respectively. The results were characterized by mean particle size, D10, D50, and D90 from the three replicates.

### 3.8. Determination of Free Fatty Acid Release Rate during the Simulated Digestion

In a process simulating intestinal digestion, the oil is hydrolyzed to free fatty acids under the catalysis of pancreatic lipase. Referring to the previous research with some modifications [18], oleic acid was added to prepare 2.5 mmol/L oleic acid-ethanol solution. The oleic acid solution was diluted with a neutral 95% ethanol solution to obtain a series of concentrations (0–400 μmol/L) of oleic acid standard solution, and titrated with 0.25 mol/L NaOH standard solution to pH neutral. A standard curve was drawn with the amount of NaOH solution consumed as the abscissa and the concentration of oleic acid as the ordinate. The release of free fatty acids during intestinal digestion was calculated according to the standard curve.

### 3.9. Observation of Micromorphology of Emulsion during Simulated Digestion

Confocal laser scanning microscope (TCS SP8, Leica, Germany) was used to observe the micromorphology of droplets in the original emulsion sample, the sample after simulated oral digestion, the sample after simulated gastric digestion, and the sample after simulated intestinal digestion for 5 min and 120 min. Nile red solution (0.01 g/mL) and Nile blue solution (0.01 g/mL) were prepared in propanol, then vortex-mixed for 30 s, respectively. 1 mL sample was mixed with 400 μL Nile red and 400 μL Nile blue solution to label the oil and starch particles on the surface of the droplet, respectively. After staining for 40 min, 10 μL emulsion drops were placed on the slide, and the cover slide was covered for observation and image collection. CLSM observation was carried out at room temperature using Ar/Kr and He/Ne visible light laser single mode, with laser wavelengths of 488 nm and 638 nm.

### 3.10. Statistical Analysis

The data was expressed as mean ± SD. The analysis of variance model (ANOVA) was used to test significant difference between means at *p* < 0.05. All statistical analyses and figures were conducted using Origin 2018 (Microcal Software, Inc., Northampton, MA, USA).

## 4. Conclusions

In summary, an emulsion-based delivery system of curcumin was successfully developed using ultrasonic treatments of rice starch-OSA as the carrier. The particle size and potential charges of OSA rice starch emulsion assisted by ultrasound were smaller than native rice starch. The particle size and potential charges did not change during in vitro oral digestion. The addition of UORS-150W and UROS-600W in the curcumin potentially increased the bioavailability of curcumin and the rate of release of fatty acids during in vitro digestion. These results have important implications for the design of emulsion-based delivery systems, which enhance the health benefits of curcumin. However, the loading function needs to be verified by in vivo experiments.

## Figures and Tables

**Figure 1 molecules-27-06955-f001:**
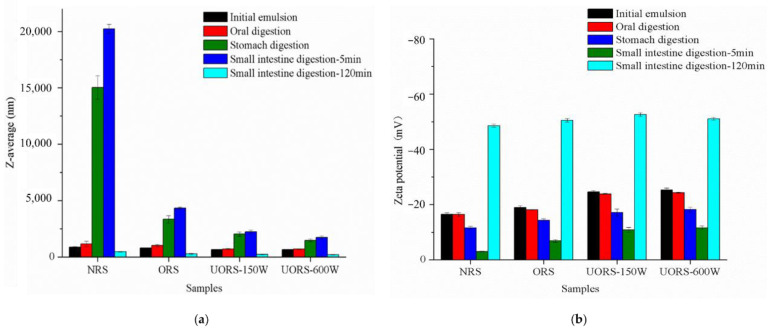
Particle size variation (**a**) and Zeta potential variation (**b**) of curcumin emulsions stabilized by native rice starch and OSA rice starch during in vitro digestion.

**Figure 2 molecules-27-06955-f002:**
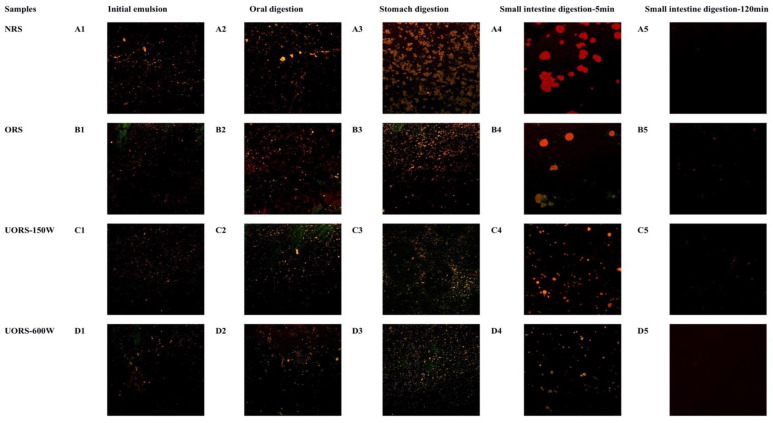
Changes in the microstructure of curcumin emulsions stabilized by native rice starch or OSA starch during in vitro digestion.

**Figure 3 molecules-27-06955-f003:**
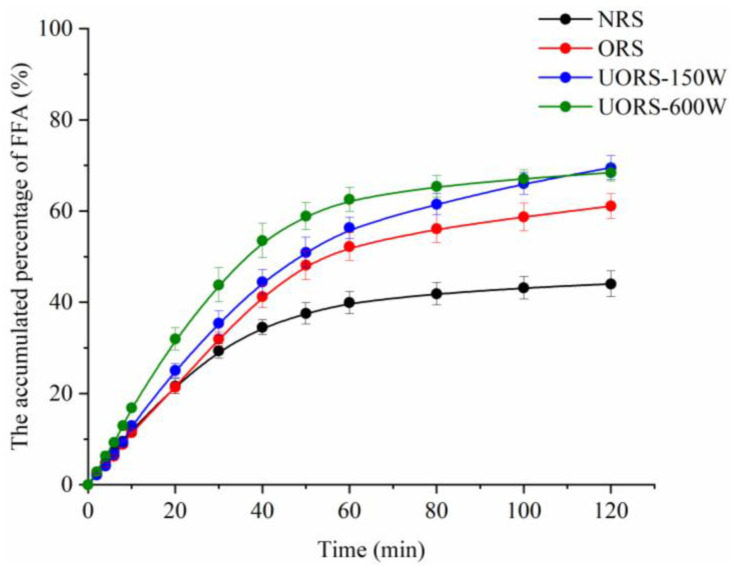
The accumulated percentage of free fatty acids (FFA%) released by curcumin emulsions stabilized by native rice starch and OSA rice starch in SIF.

**Table 1 molecules-27-06955-t001:** Encapsulation efficiency/loading capacity of curcumin in emulsions stabilized by native rice starch and OSA rice starch.

Sample	Encapsulation Efficiency (%)	Loading Capacity (µg/mg)	DS	RS
NRS	41.41 ± 0.82 ^a^	1.36 ± 0.35 ^a^	nd	nd
ORS	81.65 ± 0.14 ^b^	1.69 ± 0.22 ^b^	0.0106 ± 0.0002 ^a^	45.98 ± 0.81 ^a^
UORS-150W	85.79 ± 0.06 ^c^	1.75 ± 0.37 ^bc^	0.0113 ± 0.0005 ^a^	48.64 ± 2.02 ^a^
UORS-600W	89.03 ± 0.09 ^d^	1.81 ± 0.24 ^c^	0.0131 ± 0.0001 ^b^	56.69 ± 0.64 ^b^

Note: Different letters at the end of the same column of data represent significant differences (*p* < 0.05). nd means no data.

**Table 2 molecules-27-06955-t002:** The bioavailability of curcumin in emulsions stabilized by native rice starch and OSA rice starch.

Sample	Bioavailability of Curcumin (%)
NRS	12.58 ± 1.04 ^a^
ORS	21.63 ± 0.83 ^b^
UORS-150W	24.44 ± 0.70 ^c^
UORS-600W	24.73 ± 0.48 ^c^

Note: Different letters at the end of the same column of data represent significant differences (*p* < 0.05).

## Data Availability

Not applicable.

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
