# Peer review of "Application of Curcumin Emulsion Carrier from Ultrasonic-Assisted Prepared Octenyl Succinic Anhydride Rice Starch"

_molecules, 2022, doi:10.3390/molecules27206955_

Round 1

Reviewer 1 Report

The manuscript “Application of curcumin emulsion carrier from ultrasonic-assisted prepared octenyl succinic anhydride rice starch (molecules-1913230)”

Abstract

Line 22: Please add the expression (p>0.05) (if the statistical significance is checked at 95%) after the sentence “…..did not change significant”

Introduction

Page 2, Lines 60-62: Any findings on other bioactives (if not for curcumin)? How the ultrasonic pre-treatment affects them? As a general comment, please add more references on the studies done in encapsulation systems with the ultrasonic pre-treatment.

Page 2, Line 64: “UORS-150W and UORS-600W”, the sample abbreviations here, look like brand commercial names. This sentence should be revised.

Besides, the last paragraph (lines 63-75), should be rewritten completely. Rather than a summary of your methodology, I believe you should emphasize the novelty of the present study and it is importance in terms of its contribution to the current literature.

Results

Page 2, line 79: Please add the explanation for DS and RE, in the first use.

Pages 2-3, Lines 88-95: Can you refer to literature for this explanation?

Page 5, Lines 188-190: “At this stage, the large amount of negative charge  in the emulsion system was mainly derived from cholate, phosphoric acid, carboxyl group  (-COO-) on OSA starch, and free fatty acids released during digestion [20].” Please revise this sentence and the next one, it is not clear if this finding is related to your research or how related to your findings.

Are there any correlations with the particle size during the in vitro digestion and Zeta potential variation? Does the Figure 1a and Table 2 depict the same findings? If so, maybe you can remove the Tables completely. Is that the same for Figure 1b and Table 3? If not, please explain.

The Materials and Methods Section is clear and well-written generally. I have just one point.

Materials and Methods

Page 9, Line 342: Have you used any heat treatment for gelatinization? Please give more details.

Statistical analysis

Please mention the statistical significance level.

Table 1

Please write the explanation for RS and DS.

Author Response

The manuscript “Application of curcumin emulsion carrier from ultrasonic-assisted prepared octenyl succinic anhydride rice starch (molecules-1913230)”

Abstract

Q1: Line 22: Please add the expression (p>0.05) (if the statistical significance is checked at 95%) after the sentence “…..did not change significant”

Response: The expression (p>0.05) in the revised paper has been added as you suggested.

Introduction.

Q2: Page 2, Lines 60-62: Any findings on other bioactives (if not for curcumin)? How the ultrasonic pre-treatment affects them? As a general comment, please add more references on the studies done in encapsulation systems with the ultrasonic pre-treatment.

Response: As mention in Line 52-56, the resveratrol encapsulate was provided in the revised paper. ultrasound influences the micro-structure of starch and make it easier to interact with chemicals.

Q3: Page 2, Line 64: “UORS-150W and UORS-600W”, the sample abbreviations here, look like brand commercial names. This sentence should be revised.

Response: The abbreviation of sample was according to the name. and this sentence was revised in the paper as your suggestion.

Q4: Besides, the last paragraph (lines 63-75), should be rewritten completely. Rather than a summary of your methodology, I believe you should emphasize the novelty of the present study and it is importance in terms of its contribution to the current literature.

Response: Thanks for your suggestion, we have revised this parts as suggested and point out the novelty of the present work.

Results

Q5: Page 2, line 79: Please add the explanation for DS and RE, in the first use.

Response: Thanks for your suggestion, we have added the explanation in the revised paper.

Q6: Pages 2-3, Lines 88-95: Can you refer to literature for this explanation?

Response: the proper references were cited in the revised paper as your suggestion.

Q7: Page 5, Lines 188-190: “At this stage, the large amount of negative charge in the emulsion system was mainly derived from cholate, phosphoric acid, carboxyl group  (-COO-) on OSA starch, and free fatty acids released during digestion [20].” Please revise this sentence and the next one, it is not clear if this finding is related to your research or how related to your findings.

Response: This part has been modified carefully in the revised paper to make it much easier to understand.

Q8: Are there any correlations with the particle size during the in vitro digestion and Zeta potential variation? Does the Figure 1a and Table 2 depict the same findings? If so, maybe you can remove the Tables completely. Is that the same for Figure 1b and Table 3? If not, please explain.

Response: there was no correlation with the particle size during the in vitro digestion and Zeta potential variation, both of them could reflect the state of emulsion during digestive process. The date of Table 2 and Table 3 was the same with Figure 1a and Figure 1b, and the tables were removed in the revised paper.

The Materials and Methods Section is clear and well-written generally. I have just one point.

Materials and Methods

Q9: Page 9, Line 342: Have you used any heat treatment for gelatinization? Please give more details.

Response: We have not used heat treatment for gelatinization, all ultrasonic treatment was performed at room temperature.

Statistical analysis

Q10: Please mention the statistical significance level.

Response: The statistical significance level has been added in the revised paper.

Table 1

Q11: Please write the explanation for RS and DS.

Response: The explanation for RS and DS has been added in the revised paper.

Reviewer 2 Report

In this paper, the author attends to investigate ultrasonic assisted curcumin@OSA emulsion system, especially focusing on the in vitro properties. Although some significant result was provided, but it’s still lack of evidence to support their conclusion as standard for this journal at present. Before acceptance, additional data need to be provided.

1. please characterize OSA sample, e.g. its particle size, zeta-potential, FTIR, and morphology result

2. besides EE, the loading capacity of the prepared sample also needs to be provided as a comparison reference

3. CLSM image is hard to catch, please provide full resolutions in the supplement.

4.  system temperature during ultrasonic processing needs to be reported

5.  Many research reported UT assistant curcumin@OSA starch system, including in vitro properties, please highlight your novelth

Author Response

Reviewer 2

In this paper, the author attends to investigate ultrasonic assisted curcumin@OSA emulsion system, especially focusing on the in vitro properties. Although some significant result was provided, but it’s still lacks of evidence to support their conclusion as standard for this journal at present. Before acceptance, additional data need to be provided.

Q1: 1. please characterize OSA sample, e.g. its particle size, zeta-potential, FTIR, and morphology result

Response: As mentioned in Figure 1a and 1b, the initial particle size of OSA samples were 865.90 ± 49.20, 814.10 ± 8.34, 660.35 ± 19.87, 654.15 ± 15.63 nm, respectively. the initial zeta-potential were -16.50 ± 0.57, -18.95 ± 0.64, -24.60 ± 0.57 and -25.30 ± 0.71 mV, respectively.

Q2: besides EE, the loading capacity of the prepared sample also needs to be provided as a comparison reference

Response: The loading capacity has been added in the revised paper.

Q3: CLSM image is hard to catch, please provide full resolutions in the supplement.

Response: The full resolutions of CLSM images have been provided in the supplement as you suggested

Q4: system temperature during ultrasonic processing needs to be reported

Response: As according to Yang et al.(2019), the temperature of starch slurry was 25 ℃ during ultrasonic processing. The detailed information was provided in the revised paper.

Q5: Many researchers reported UT assistant curcumin@OSA starch system, including in vitro properties, please highlight your novelthy

Response: We know that many researchers reported UT assistant curcumin@OSA starch system, however, seldom work has focused on this work systematically. In present study, we try to use ultrasonic-assisted OSA rice starch as an emulsifier to construct curcumin emulsion carrier. the loading rate/capacity and bioavailability of different emulsions were measured, the particle size, Zeta potential, apparent morphology and free fatty acid release of the emulsion at different in vitro digestive stages were observed, and finally the the releasing mechanism of ultrasonic-assisted OSA rice starch during in vitro digestion were analyzed. As a systematical work, our study might provide reference for the development of potential application of ultrasonic-assisted OSA rice starch on encapsulation of bio actives.

Reviewer 3 Report

The paper presented for review entitled “Aplication of curcumin emulsion carrier from ultra-assisted prepared octenyl succinic anhydride rice starch“ offers practical and useful finding that ultra-assisted prepared octenyl succinic anhydride rice starch could improve curcumin bioavailability and proposes a new way to develop an emulsion-based delivery system of curcumin.

The analyzes that covered the subject of this paper are comprehensive, and the results are well explained and supported by relevant literature. The manuscript is generally well written, but several issues require correction or clarification, as per the comments below:

Line 34-51 Add more references for these statements.

Line 57 The number of given reference is missing.

To make it easier for the reader to follow the results and discussion, introduce the material and methods section immediately after the introduction.

Line 111-124 Support facts, conclusions and assumptions with appropriate literature references.

Tn the conclusion, mention the results of in vitro oral digestion and the author's proposal for future research on this topic.

Line 319 Correct as samples and chemicals.

Line 327-329 Better explain ultrasound treatment and the equipment used.

For the following devices provide information about the manufacturer, city and country,: centrifuge, oven, disperser, homogenizer

Line 352-353 With which the absorbance was measured? Provide information about the spectrophotometer used.

Line 413-414 Do not use the imperative in the methods description.

Author Response

Reviewer 3

The paper presented for review entitled “Aplication of curcumin emulsion carrier from ultra-assisted prepared octenyl succinic anhydride rice starch “offers practical and useful finding that ultra-assisted prepared octenyl succinic anhydride rice starch could improve curcumin bioavailability and proposes a new way to develop an emulsion-based delivery system of curcumin.

The analyzes that covered the subject of this paper are comprehensive, and the results are well explained and supported by relevant literature. The manuscript is generally well written, but several issues require correction or clarification, as per the comments below:

Q1: Line 34-51 Add more references for these statements.

Response: The proper references were cited in the revised paper as you suggested.

Q2: Line 57 The number of given references is missing.

Response: Thanks for your suggestion, the number of the reference was added in the revised paper.

Q3: To make it easier for the reader to follow the results and discussion, introduce the material and methods section immediately after the introduction.

Response: Thanks for your suggestion and the layout of this manuscript was based on the requirement of this journal, so we follow the demand which first was results and discussion and then materials and methods.

Q4: Line 111-124 Support facts, conclusions and assumptions with appropriate literature references.

Response: Several references were supplied in the revised paper as your suggestion.

Q5: Tn the conclusion, mention the results of in vitro oral digestion and the author's proposal for future research on this topic.

Response: thanks for your suggestion, the results of in vitro oral digestion and the future proposal was listed in the revised paper.

Q6: Line 319 Correct as samples and chemicals.

Response: We have corrected it as your suggestion in the revised paper.

Q7: Line 327-329 Better explain ultrasound treatment and the equipment used.

Response: Detailed information of ultrasonic treatment was provided in the revised paper.

Q8: For the following devices provide information about the manufacturer, city and country: centrifuge, oven, disperser, homogenizer

Response: Thanks for your suggestion, we have provided the detailed information of devices used.

Q9: Line 352-353 With which the absorbance was measured? Provide information about the spectrophotometer used.

Response: The absorbance of curcumin solution was measured, and the information of spectrophotometer was provided in the revised paper

Q10: Line 413-414 Do not use the imperative in the methods description.

Response: Thanks for your suggestion, we have corrected this sentence I the revised paper.

Round 2

Reviewer 1 Report

Thanks for the revisions. I believe they improved the manuscript more.

Reviewer 2 Report

All comments have been addressed